# Graph dynamical networks for unsupervised learning of atomic scale dynamics in materials

Tian Xie[1], Arthur France-Lanord[1], Yanming Wang [1], Yang Shao-Horn[2] & Jeffrey C. Grossman [1]

Understanding the dynamical processes that govern the performance of functional materials is essential for the design of next generation materials to tackle global energy and environmental challenges. Many of these processes involve the dynamics of individual atoms or small molecules in condensed phases, e.g. lithium ions in electrolytes, water molecules in membranes, molten atoms at interfaces, etc., which are difficult to understand due to the complexity of local environments. In this work, we develop graph dynamical networks, an unsupervised learning approach for understanding atomic scale dynamics in arbitrary phases and environments from molecular dynamics simulations. We show that important dynamical information, which would be difficult to obtain otherwise, can be learned for various multi-component amorphous material systems. With the large amounts of molecular dynamics data generated every day in nearly every aspect of materials design, this approach provides a broadly applicable, automated tool to understand atomic scale dynamics in material systems.

[1] Department of Materials Science and Engineering, Massachusetts Institute of Technology, Cambridge, MA 02139, USA. [2] Department of Mechanical Engineering, Massachusetts Institute of Technology, Cambridge, MA 02139, USA. Correspondence and requests for materials should be addressed to J.C.G. (email: jcg@mit.edu)

Understanding the atomic scale dynamics in condensed phases is essential for the design of functional materials to tackle global energy and environmental challenges[1–3]. The performance of many materials depends on the dynamics of individual atoms or small molecules in complex local environments. Despite the rapid advances in experimental techniques[4–6], molecular dynamics (MD) simulations remain one of the few tools for probing these dynamical processes with both atomic scale time and spatial resolutions. However, due to the large amounts of data generated in each MD simulation, it is often challenging to extract statistically relevant dynamics for each atom especially in multi-component, amorphous material systems. At present, atomic scale dynamics are usually learned by designing system-specific descriptions of coordination environments or computing the average behavior of atoms[7–10]. A general approach for understanding the dynamics in different types of condensed phases, including solid, liquid, and amorphous, is still lacking.

The advances in applying deep learning to scientific research open new opportunities for utilizing the full trajectory data from MD simulations in an automated fashion. Ideally, one would trace every atom or small molecule of interest in the MD trajectories, and summarize their dynamics into a linear, low dimensional model that describes how their local environments evolve over time. Recent studies show that combining Koopman analysis and deep neural networks provides a powerful tool to understand complex biological processes and fluid dynamics from data[11–13]. In particular, VAMPnets[13] develop a variational approach for Markov processes to learn an optimal latent space representation that encodes the long-time dynamics, which enables the end-to-end learning of a linear dynamical model directly from MD data. However, in order to learn the atomic dynamics in complex, multi-component material systems, sharing knowledge learned for similar local chemical environments is essential to reduce the amount of data needed. The recent development of graph convolutional neural networks (GCN) has led to a series of new representations of molecules[14–17] and materials[18,19] that are invariant to permutation and rotation operations. These representations provide a general approach to encode the chemical structures in neural networks which shares parameters between different local environments, and they have been used for predicting properties of molecules and materials[14–19], generating force fields[19,20], and visualizing structural similarities[21,22].

In this work, we develop a deep learning architecture, Graph Dynamical Networks (GDyNets), that combines Koopman analysis and graph convolutional neural networks to learn the dynamics of individual atoms in material systems. The graph convolutional neural networks allow for the sharing of knowledge learned for similar local environments across the system, and the variational loss developed in VAMPnets[13,23] is employed to learn a linear model for atomic dynamics. Thus, our method focuses on the modeling of local atomic dynamics instead of global dynamics. This significantly improves the sampling of the atomic dynamical processes, because a typical material system includes a large number of atoms or small molecules moving in structurally similar but distinct local environments. We demonstrate this distinction using a toy system that shows global dynamics can be exponentially more complex than local dynamics. Then, we apply this method to two realistic material systems—silicon dynamics at solid–liquid interfaces and lithium ion transport in amorphous polymer electrolytes—to demonstrate the new dynamical information one can extract for such complex materials and environments. Given the enormous amount of MD data generated in nearly every aspect of materials research, we believe the broad applicability of this method could help uncover important new

physical insights from atomic scale dynamics that may have otherwise been overlooked.

## Results

**Koopman analysis of atomic scale dynamics.** In materials design, the dynamics of target atoms, like the lithium ion in electrolytes and the water molecule in membranes, provide key information to material performance. We describe the dynamics of the target atoms and their surrounding atoms as a discrete process in MD simulations,

$$\boldsymbol{x}_{t+\tau} = \boldsymbol{F}(\boldsymbol{x}_t), \tag{1}$$

where $\boldsymbol{x}_t$ and $\boldsymbol{x}_{t+\tau}$ denote the local configuration of the target atoms and their surrounding atoms at time steps $t$ and $t + \tau$, respectively. Note that Eq. (1) implies that the dynamics of $\boldsymbol{x}$ is Markovian, i.e. $\boldsymbol{x}_{t+\tau}$ only depends on $\boldsymbol{x}_t$ not the configurations before it. This is exact when $\boldsymbol{x}$ includes all atoms in the system, but an approximation if only neighbor atoms are included. We also assume that each set of target atoms follow the same dynamics $\boldsymbol{F}$. These are valid assumptions since (1) most interactions in materials are short-range, (2) most materials are either periodic or have similar local structures, and we could test them by validating the dynamical models using new MD data, which we will discuss later.

The Koopman theory[24] states that there exists a function $\chi(\boldsymbol{x})$ that maps the local configuration of target atoms $\boldsymbol{x}$ into a lower dimensional feature space, such that the non-linear dynamics $\boldsymbol{F}$ can be approximated by a linear transition matrix $\boldsymbol{K}$,

$$\chi(\boldsymbol{x}_{t+\tau}) \approx \boldsymbol{K}^T \chi(\boldsymbol{x}_t). \tag{2}$$

The approximation becomes exact when the feature space has infinite dimensions. However, for most dynamics in material systems, it is possible to approximate it with a low dimensional feature space if $\tau$ is sufficiently large due to the existence of characteristic slow processes. The goal is to identify such slow processes by finding the feature map function $\chi(\boldsymbol{x})$.

**Learning feature map function with graph dynamical networks.** In this work, we use GCN to learn the feature map function $\chi(\boldsymbol{x})$. GCN provides a general framework to encode the structure of materials that is invariant to permutation, rotation, and reflection[18,19]. As shown in Fig. 1, for each time step in the MD trajectory, a graph $\mathcal{G}$ is constructed based on its current configuration with each node $v_i$ representing an atom and each edge $\boldsymbol{u}_{i,j}$ representing a bond connecting nearby atoms. We connect $M$ nearest neighbors considering periodic boundary conditions while constructing the graph, and a gated architecture[18] is used in GCN to reweigh the strength of each connection (see Supplementary Note 1 for details). Note that the graphs are constructed separately for each step, so the topology of each graph may be different. Also, the 3-dimensional information is preserved in the graphs since the bond length is encoded in $\boldsymbol{u}_{i,j}$. Then, each graph is input to the same GCN to learn an embedding for each atom through graph convolution (or neural message passing[16]) that incorporates the information of its surrounding environments.

$$\boldsymbol{v}_i' = \mathrm{Conv}(\boldsymbol{v}_i, \boldsymbol{v}_j, \boldsymbol{u}_{(i,j)}), \quad (i,j) \in \mathcal{G}. \tag{3}$$

After $K$ convolution operations, information from the $K$th neighbors will be propagated to each atom, resulting in an embedding $\boldsymbol{v}_i^{(K)}$ that encodes its local environment.

To learn a feature map function for the target atoms whose dynamics we want to model, we focus on the embeddings learned for these atoms. Assume that there are $n$ sets of target atoms each made up with $k$ atoms in the material system. For instance, in a

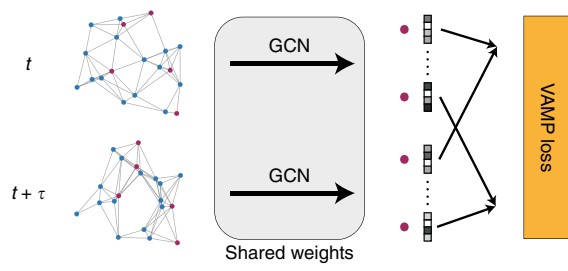

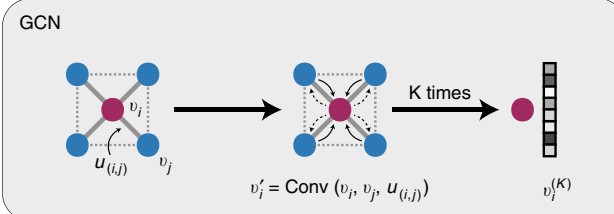

**Fig. 1** Illustration of the graph dynamical networks architecture. The MD trajectories are represented by a series of graphs dynamically constructed at each time step. The red nodes denote the target atoms whose dynamics we are interested in, and the blue nodes denote the rest of the atoms. The graphs are input to the same graph convolutional neural network to learn an embedding $\mathbf{v}_i^{(K)}$ for each atom that represents its local configuration. The embeddings of the target atoms at $t$ and $t + \tau$ are merged to compute a VAMP loss that minimizes the errors in Eq. (2)

system of 10 water molecules, $n = 10$ and $k = 3$. We use the label $\mathbf{v}_{[l,m]}$ to denote the $m$th atom in the $l$th set of target atoms. With a pooling function[18], we can get an overall embedding $\mathbf{v}_{[l]}$ for each set of target atoms to represent its local configuration,

$$\mathbf{v}_{[l]} = \mathrm{Pool}(\mathbf{v}_{[l,0]}, \mathbf{v}_{[l,1]}, \ldots, \mathbf{v}_{[l,k]}). \quad (4)$$

Finally, we build a shared two-layer fully connected neural network with an output layer using a Softmax activation function to map the embeddings $\mathbf{v}_{[l]}$ to a feature space $\widetilde{\mathbf{v}}_{[l]}$ with a pre-determined dimension. This is the feature space described in Eq. (2), and we can select an appropriate dimension to capture the important dynamics in the material system. The Softmax function used here allows us to interpret the feature space as a probability over several states[13]. Below, we will use the term "number of states" and "dimension of feature space" interchangeably.

To minimize the errors of the approximation in Eq. (2), we compute the loss of the system using a VAMP-2 score[13,24] that measures the consistency between the feature vectors learned at timesteps $t$ and $t + \tau$,

$$\mathrm{Loss} = -\mathrm{VAMP}(\widetilde{\mathbf{v}}_{[l],t}, \widetilde{\mathbf{v}}_{[l],t+\tau}), \quad t \in [0, T - \tau], l \in [0, n]. \quad (5)$$

This means that a single VAMP-2 score is computed over the whole trajectory and all sets of target atoms. The entire network is trained by minimizing the VAMP loss, i.e. maximizing the VAMP-2 score, with the trajectories from the MD simulations.

**Hyperparameter optimization and model validation.** There are several hyperparameters in the GDyNets that need to be optimized, including the architecture of GCN, the dimension of the feature space, and lag time $\tau$. We divide the MD trajectory into training, validation, and testing sets. The models are trained with trajectories from the training set, and a VAMP-2 score is computed with trajectories from the validation set. The GCN architecture is optimized according to the VAMP-2 score similar to ref. [18].

The accuracy of Eq. (2) can be evaluated with a Chapman-Kolmogorov (CK) equation,

$$\mathbf{K}(n\tau) = \mathbf{K}^n(\tau), \quad n = 1, 2, \ldots . \quad (6)$$

This equation holds if the dynamic model learned is Markovian, and it can predict the long-time dynamics of the system. In general, increasing the dimension of feature space makes the dynamic model more accurate, but it may result in overfitting when the dimension is very large. Since a higher feature space dimension and a larger $\tau$ make the model harder to understand and contain less dynamical details, we select the smallest feature space dimension and $\tau$ that fulfills the CK equation within statistical uncertainty. Therefore, the resulting model is interpretable and contains more dynamical details. Further details regarding the effects of feature space dimension and $\tau$ can be found in refs. [13,24].

**Local and global dynamics in the toy system.** To demonstrate the advantage of learning local dynamics in material systems, we compare the dynamics learned by the GDyNet with VAMP loss and a standard VAMPnet with fully connected neural networks that learns global dynamics for a simple model system using the same input data. As shown in Fig. 2a, we generated a 200 ns MD trajectory of a lithium atom moving in a face-centered cubic (FCC) lattice of sulfur atoms at a constant temperature, which describes an important lithium ion transport mechanism in solid-state electrolytes[7]. There are two different sites for the lithium atom to occupy in a FCC lattice, tetrahedral sites and octahedral sites, and the hopping between the two sites should be the only dynamics in this system. As shown in Fig. 2b–d, after training and validation with the first 100 ns trajectory, the GDyNet correctly identified the transition between the two sites with a relaxation timescale of 42.3 ps while testing on the second 100 ns trajectory, and it performs well in the CK test. In contrast, the standard VAMPnet, which inputs the same data as the GDyNet, learns a global transition with a much longer relaxation timescale at 236 ps, and it performs much worse in the CK test. This is because the model views the four octahedral sites as different sites due to their different spatial locations. As a result, the transitions between these identical sites are learned as the slowest global dynamics.

It is theoretically possible to identify the faster local dynamics from a global dynamical model when we increase the dimension of feature space (Supplementary Fig. 1). However, when the size of the system increases, the number of slower global transitions will increase exponentially, making it practically impossible to discover important atomic scale dynamics within a reasonable simulation time. In addition, it is possible in this simple system to design a symmetrically invariant coordinate to include the equivalence of the octahedral and tetrahedral sites. But in a more complicated multi-component or amorphous material system, it is difficult to design such coordinates that take into account the complex atomic local environments. Finally, it is also possible to reconstruct global dynamics from the local dynamics. Since we know how the four octahedral and eight tetrahedral sites are connected in a FCC lattice, we can construct the 12 dimensional global transition matrix from the 2 dimensional local transition matrix (see Supplementary Note 2 for details). We obtain the slowest global relaxation timescale to be 531 ps, which is close to the observed slowest timescale of 528 ps from the global dynamical model in Supplementary Fig. 1. Note that the timescale from the two-state global model in Fig. 2 is less accurate since it fails to learn the correct transition. In sum, the built-in invariances in GCN provide a general approach to reduce the complexity of learning atomic dynamics in material systems.

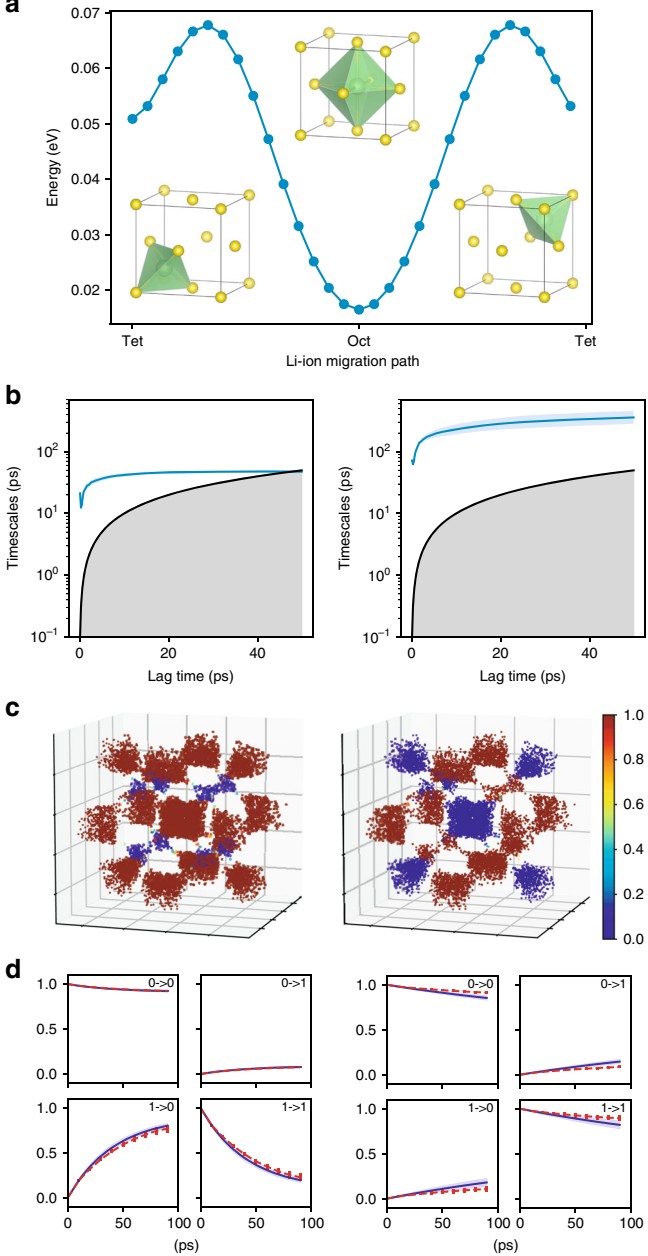

**Fig. 2** A two-state dynamic model learned for a lithium ion in the face-centered cubic lattice. **a** Structure of the FCC lattice and the relative energies of the tetrahedral and octahedral sites. **b**–**d** Comparison between the local dynamics (left) learned with GDyNet and the global dynamics (right) learned with a standard VAMPnet. **b** Relaxation timescales computed from the Koopman models as a function of the lag time. The black lines are reference lines where the relaxation timescale equals to the lag time. **c** Assignment of the two states in the FCC lattice. The color denotes the probability of being in state 0, which corresponds to one of the two states that has a larger population. **d** CK test comparing the long-time dynamics predicted by Koopman models at $\tau = 10$ ps (blue) and actual dynamics (red). The shaded areas and error bars in **b**, **d** report the 95% confidence interval from five independent trajectories by dividing the test data equally into chunks

**Silicon dynamics at a solid–liquid interface**. To evaluate the performance of the GDyNets with VAMP loss for a more complicated system, we study the dynamics of silicon atoms at a binary solid–liquid interface. Understanding the dynamics at interfaces is notoriously difficult due to the complex local

structures formed during phase transitions[25,26]. As shown in Fig. 3a, an equilibrium system made of two crystalline Si {110} surfaces and a liquid Si–Au solution is constructed at the eutectic point (629 K, 23.4% Si[27]) and simulated for 25 ns using MD. We train and validate a four-state model using the first 12.5 ns trajectory, and use it to identify the dynamics of Si atoms in the last 12.5 ns trajectory. Note that we only use the Si atoms in the liquid phase and the first two layers of the solid {110} surfaces as the target atoms (Fig. 3b). This is because the Koopman models are optimized for finding the slowest transition in the system, and including additional solid Si atoms will result in a model that learns the slower Si hopping in the solid phase which is not our focus.

In Fig. 3b, c, the model identified four states that are crucial for the Si dynamics at the solid–liquid interface – liquid Si at the interface (state 0), solid Si (state 1), solid Si at the interface (state 2), and liquid Si (state 3). These states provide a more detailed description of the solid–liquid interface structure than conventional methods. In Supplementary Fig. 2, we compare our results with the distribution of the $q_3$ order parameter of the Si atoms in the system, which measures how much a site deviates from a diamond-like structure and is often used for studying Si interfaces[28]. We learn from the comparison that (1) our method successfully identifies the bulk liquid and solid states, and learns additional interface states that cannot be obtained from $q_3$; (2) the states learned by our method are more robust due to access to dynamical information, while $q_3$ can be affected by the accidental ordered structures in the liquid phase; (3) $q_3$ is system specific and only works for diamond-like structures, but the GDyNets can potentially be applied to any material given the MD data.

In addition, important dynamical processes at the solid–liquid interface can be learned with the model. Remarkably, the model identified the relaxation process of the solid–liquid transition with a timescale of 538 ns (Fig. 3d, e), which is one order of magnitude longer than the simulation time of 12.5 ns. This is because the large number of Si atoms in the material system provide an ensemble of independent trajectories that enable the identification of rare events[29–31]. The other two relaxation processes correspond to the transitions of solid Si atoms into/out of the interface (73.2 ns) and liquid Si atoms into/out of the interface (2.26 ns), respectively. These processes are difficult to obtain with conventional methods due to the complex structures at solid–liquid interfaces, and the results are consistent with our understanding that the former solid relaxation is significantly slower than the latter liquid relaxation. Finally, the model performs excellently in the CK test on predicting the long-time dynamics.

**Lithium ion dynamics in polymer electrolytes**. Finally, we apply GDyNets with VAMP loss to study the dynamics of lithium ions (Li-ions) in solid polymer electrolytes (SPEs), an amorphous material system composed of multiple chemical species. SPEs are candidates for next-generation battery technology due to their safety, stability, and low manufacturing cost, but they suffer from low Li-ion conductivity compared with liquid electrolytes[32,33]. Understanding the key dynamics that affect the transport of Li-ions is important to the improvement of Li-ion conductivity in SPEs.

We focus on the state-of-the-art[33] SPE system—a mixture of poly(ethylene oxide) (PEO) and lithium bis-trifluoromethyl sulfonimide (LiTFSI) with Li/EO = 0.05 and a degree of polymerization of 50, as shown in Fig. 4a. Five independent 80 ns trajectories are generated to model the Li-ion transport at 363 K, following the same approach as described in ref. [67]. We train a four-state GDyNet with one of the trajectories, and use the

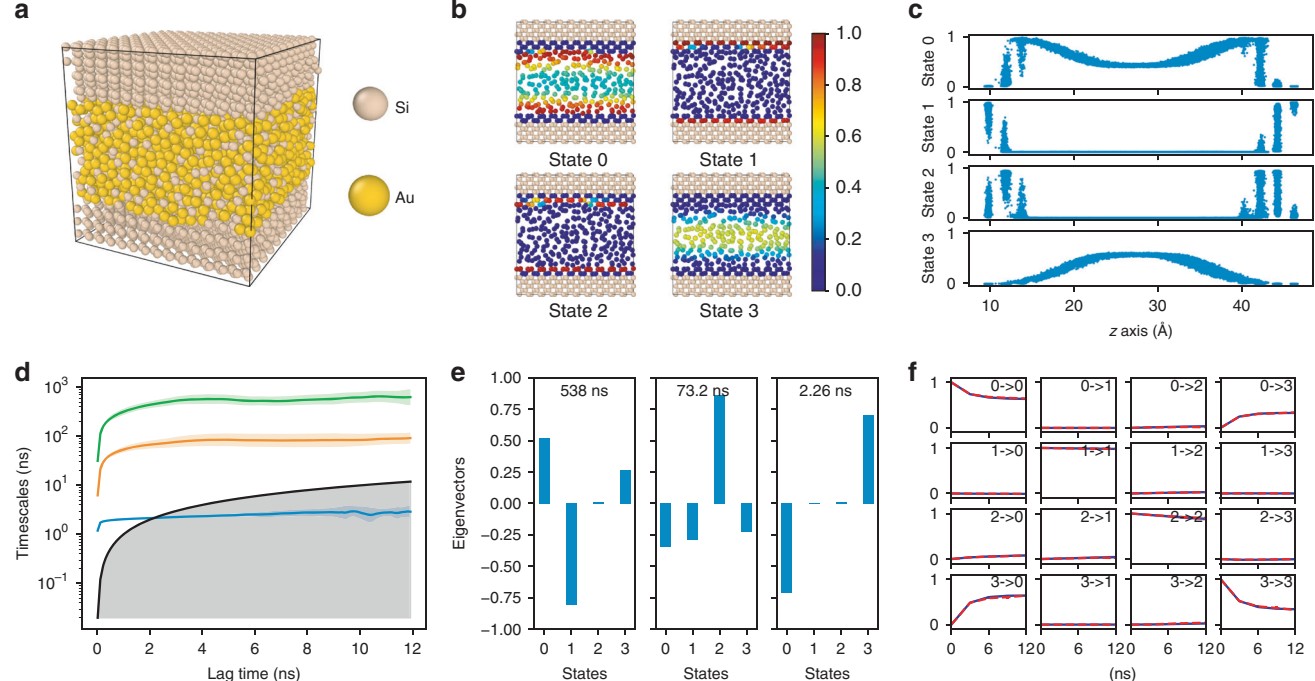

**Fig. 3** A four-state dynamical model learned for silicon atoms at a solid–liquid interface. **a** Structure of the silicon-gold two-phase system. **b** Cross section of the system, where only silicon atoms are shown and color-coded with the probability of being in each state. **c** The distribution of silicon atoms in each state as a function of $z$-axis coordinate. **d** Relaxation timescales computed from the Koopman models as a function of the lag time. The black lines are reference lines where the relaxation timescale equals to the lag time. **e** Eigenvectors projected to each state for the three relaxations of Koopman models at $\tau = 3$ ns. **f** CK test comparing the long-time dynamics predicted by Koopman models at $\tau = 3$ ns (blue) and actual dynamics (red). The shaded areas and error bars in **d**, **f** report the 95% confidence interval from five sets of Si atoms by randomly dividing the target atoms in the test data

model to identify the dynamics of Li-ions in the remaining four trajectories. The model identified four different solvation environments, i.e. states, for the Li-ions in the SPE. In Fig. 4b, the state 0 Li-ion has a population of $50.6 \pm 0.8\%$, and it is coordinated by a PEO chain on one side and a TFSI anion on the other side. The state 1 has a similar structure as state 0 with a population of $27.3 \pm 0.4\%$, but the Li-ion is coordinated by a hydroxyl group on the PEO side rather than an oxygen. In state 2, the Li-ion is completely coordinated by TFSI anion ions, which has a population of $15.1 \pm 0.4\%$. And the state 3 Li-ion is coordinated by PEO chains with a population of $7.0 \pm 0.9\%$. Note that the structures in Fig. 4b only show a representative configuration for each state. We compute the element-wise radial distribution function (RDF) for each state in Supplementary Fig. 3 to demonstrate the average configurations, which is consistent with the above description. We also analyze the total charge carried by the Li-ions in each state considering their solvation environments in Fig. 4c (see Supplementary Note 3 and Supplementary Table 1 for details). Interestingly, both state 0 and state 1 carry almost zero total charge in their first solvation shell due to the one TFSI anion in their solvation environments.

We further study the transition between the four Li-ion states. Three relaxation processes are identified in the dynamical model as shown in Fig. 4d, e. By analyzing the eigenvectors, we learn that the slowest relaxation is a process involving the transport of a Li-ion into and out of a PEO coordinated environment. The second slowest relaxation happens mainly between state 0 and state 1, corresponding to a movement of the hydroxyl end group. The transitions from state 0 to states 2 and 3 constitute the last relaxation process, as state 0 can be thought of an intermediate state between state 2 and state 3. The model performs well in CK tests (Fig. 4f). Relaxation processes in the PEO/LiTFSI systems have been extensively studied experimentally[34,35], but it is

difficult to pinpoint the exact atomic scale dynamics related to these relaxations. The dynamical model learned by GDyNet provides additional insights into the understanding of Li-ion transport in polymer electrolytes.

**Implications to lithium ion conduction**. The state configurations and dynamical model allow us to further quantify the transitions that are responsible for the Li-ion conduction. In Fig. 5, we compute the contribution from each state transition to the Li-ion conduction using the Koopman model at $\tau = 0.8$ ns. First, we learn that the majority of conduction results from transitions within the same states ($i \rightarrow i$). This is because the transport of Li-ions in PEO is strongly coupled with segmental motion of the polymer chains[8,36], in contrast to the hopping mechanism in inorganic solid electrolytes[37]. In addition, due to the low charge carried by state 0 and state 1, the majority of charge conduction results from the diffusion of states 2 and 3, despite their relatively low populations. Interestingly, the diffusion of state 2, a negatively charged species, accounts for ~40% of the Li-ion conduction. This provides an atomic scale explanation to the recently observed negative transference number at high salt concentration PEO/LiTFSI systems[38].

## Discussion

We have developed a general approach, GDyNets, to understand the atomic scale dynamics in material systems. Despite being widely used in biophysics[31], fluid dynamics[39], and kinetic modeling of chemical reactions[40–42], Koopman models, (or Markov state models[31], master equation methods[43,44]) have not been used in learning atomic scale dynamics in materials from MD simulations except for a few examples in understanding solvent dynamics[45–47]. Our approach also differs from several other unsupervised learning

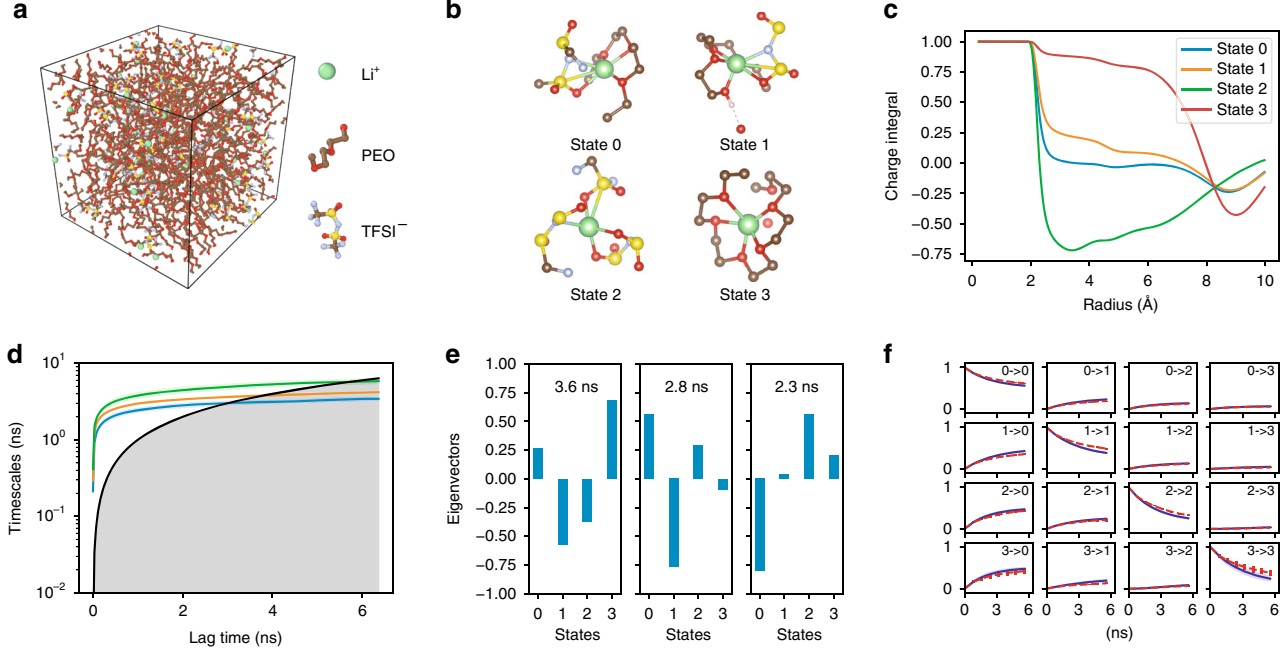

**Fig. 4** A four-state dynamical model learned for lithium ion in a PEO/LiTFSI polymer electrolyte. **a** Structure of the PEO/LiTFSI polymer electrolyte. **b** Representative configurations of the four Li-ion states learned by the dynamical model. **c** Charge integral of each state around a Li-ion as a function of radius. **d** Relaxation timescales computed from the Koopman models as a function of the lag time. The black lines are reference lines where the relaxation timescale equals to the lag time. **e** Eigenvectors projected to each state for the three relaxations of Koopman models at $\tau = 0.8$ ns. **f** CK test comparing the long-time dynamics predicted by Koopman models at $\tau = 0.8$ ns (blue) and actual dynamics (red). The shaded areas and error bars in **d**, **f** report the 95% confidence interval from four independent trajectories in the test data

methods[48–50] by directly learning a linear Koopman model from MD data. Many crucial processes that affect the performance of materials involve the local dynamics of atoms or small molecules, like the dynamics of lithium ions in battery electrolytes[51,52], the transport of water and salt ions in water desalination membranes[53,54], the adsorption of gas molecules in metal organic frameworks[55,56], among many other examples. With the improvement of computational power and continued increase in the use of molecular dynamics to study materials, this work could have broad applicability as a general framework for understanding the atomic scale dynamics from MD trajectory data.

Compared with the Koopman models previously used in biophysics and fluid dynamics, the introduction of graph convolutional neural networks enables parameter sharing between the atoms and an encoding of local environments that is invariant to permutation, rotation, and reflection. This symmetry facilitates the identification of similar local environments throughout the materials, which allows the learning of local dynamics instead of exponentially more complicated global dynamics. In addition, it is easy to extend this method to learn global dynamics with a global pooling function[18]. However, a hierarchical pooling function is potentially needed to directly learn the global dynamics of large biological systems including thousands of atoms. It is also possible to represent the local environments using other symmetry functions like smooth overlap of atomic positions (SOAP)[57], social permutation invariant (SPRINT) coordinates[58], etc. By adding a few layers of neural networks, a similar architecture can be designed to learn the local dynamics of atoms. However, these built-in invariances may also cause the Koopman model to ignore dynamics between symmetrically equivalent structures which might be important to the material performance. One simple example is the flip of an ammonia molecule—the two states are mirror symmetric to each other so the GCN will not be

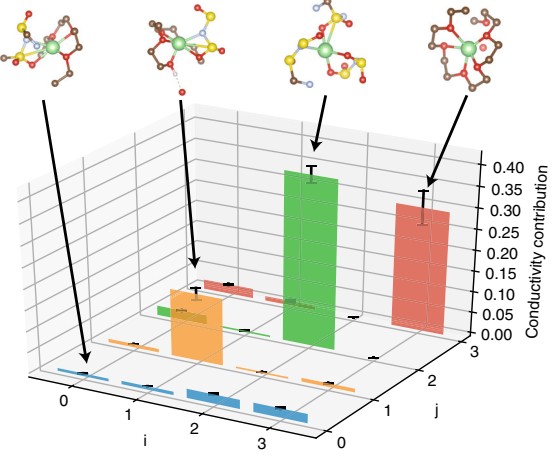

**Fig. 5** Contribution from each transition to lithium ion conduction. Each bar denotes the percentage that the transition from state *i* to state *j* contributes to the overall lithium ion conduction. The error bars report the 95% confidence interval from four independent trajectories in test data

able to differentiate them by design. This can potentially be resolved by partially breaking the symmetry of GCN based on the symmetry of the material systems.

The graph dynamical networks can be further improved by incorporating ideas from both the fields of Koopman models and graph neural networks. For instance, the auto-encoder architecture[12,59,60] and deep generative models[61] start to enable the direct generation of future structures in the configuration space. Our method currently lacks a generative component, but this can potentially be achieved with a proper graph decoder[62,63]. Furthermore, transfer learning on graph embeddings may reduce

the number of MD trajectories needed for learning the dynamics[64,65].

In summary, graph dynamical networks present a general approach for understanding the atomic scale dynamics in materials. With a toy system of lithium ion transporting in a face-centered cubic lattice, we demonstrate that learning local dynamics of atoms can be exponentially easier than global dynamics in material systems with representative local structures. The dynamics learned from two more complicated systems, solid–liquid interfaces and solid polymer electrolytes, indicate the potential of applying the method to a wide range of material systems and understanding atomic dynamics that are crucial to their performances.

## Methods

**Construction of the graphs from trajectory.** A separate graph is constructed using the configuration in each time step. Each atom in the simulation box is represented by a node $i$ whose embedding $\boldsymbol{v}_i$ is initialized randomly according to the element type. The edges are determined by connecting $M$ nearest neighbors whose embedding $\boldsymbol{u}_{(i,j)}$ is calculated by,

$$\boldsymbol{u}_{(i,j)}[t] = \exp(-(d_{(i,j)} - \mu_t)^2/\sigma^2), \tag{7}$$

where $\mu_t = t \cdot 0.2$ Å for $t = 0, 1, \dots, K$, $\sigma = 0.2$ Å, and $d_{(i,j)}$ denotes the distance between $i$ and $j$ considering the periodic boundary conditions. The number of nearest neighbors $M$ is 12, 20, and 20 for the toy system, Si–Au binary system, and PEO/LiTFSI system, respectively.

**Graph convolutional neural network architecture details.** The convolution function we employed in this work is similar to those in refs. [18,22] but features an attention layer[66]. For each node $i$, we first concatenate neighbor vectors from the last iteration $\boldsymbol{z}_{(i,j)}^{(t-1)} = \boldsymbol{v}_i^{(t-1)} \oplus \boldsymbol{v}_j^{(t-1)} \oplus \boldsymbol{u}_{(i,j)}$, then we compute the attention coefficient of each neighbor,

$$\alpha_{ij} = \frac{\exp(\boldsymbol{z}_{(i,j)}^{(t-1)} \boldsymbol{W}_a^{(t-1)} + b_a^{(t-1)})}{\sum_j \exp(\boldsymbol{z}_{(i,j)}^{(t-1)} \boldsymbol{W}_a^{(t-1)} + b_a^{(t-1)})}, \tag{8}$$

where $\boldsymbol{W}_a^{(t-1)}$ and $b_a^{(t-1)}$ denotes the weights and biases of the attention layers and the output $\alpha_{ij}$ is a scalar number between 0 and 1. Finally, we compute the embedding of node $i$ by,

$$\boldsymbol{v}_i^{(t)} = \boldsymbol{v}_i^{(t-1)} + \sum_j \alpha_{ij} \cdot g(\boldsymbol{z}_{(i,j)}^{(t-1)} \boldsymbol{W}_n^{(t-1)} + \boldsymbol{b}_n^{(t-1)}), \tag{9}$$

where $g$ denotes a non-linear ReLU activation function, and $\boldsymbol{W}_n^{(t-1)}$ and $\boldsymbol{b}_n^{(t-1)}$ denotes weights and biases in the network.

The pooling function computes the average of the embeddings of each atom for the set of target atoms,

$$\boldsymbol{v}_{[l]} = \frac{1}{k} \sum_m \boldsymbol{v}_{[l,m]}. \tag{10}$$

**Determination of the relaxation timescales.** The relaxation timescales represent the characteristic timescales implied by the transition matrix $\boldsymbol{K}(\tau)$, where $\tau$ denotes the lag time of the transition matrix. By conducting an eigenvalue decomposition for $\boldsymbol{K}(\tau)$, we could compute the relaxation timescales as a function of lag time by,

$$t_i(\tau) = -\frac{\tau}{\ln|\lambda_i(\tau)|}, \tag{11}$$

where $\lambda_i(\tau)$ denotes the $i$th eigenvalue of the transition matrix $\boldsymbol{K}$. Note that the largest eigenvalue is alway 1, corresponding to infinite relaxation timescale and the equilibrium distribution. The finite $t_i(\tau)$ are plotted in Figs. 2b, 3d, and 4d for each material system as a function of $\tau$ by performing this computation using the corresponding $\boldsymbol{K}(\tau)$. If the dynamics of the system is Markovian, i.e. Eq. (6) holds, one can prove that the relaxation timescales $t_i(\tau)$ will be constant for any $\tau$[13,24]. Therefore, we select a smallest $\tau^*$ from Figs. 2b, 3d, and 4d to obtain a dynamical model that is Markovian and contains most dynamical details. We then compute the relaxation timescales using this $\tau^*$ for each material system, and these timescales remain constant for any $\tau > \tau^*$.

**State-weighted radial distribution function.** The RDF describes how particle density varies as a function of distance from a reference particle. The RDF is usually determined by counting the neighbor atoms at different distances over MD trajectories. We calculate the RDF of each state by weighting the counting process

according to the probability of the reference particle being in state $i$,

$$g_i(r_A) = \frac{1}{\rho_i} \frac{\mathrm{d}[n(r_A) \cdot p_i]}{4\pi r_A^2 \mathrm{d}r_A}, \tag{12}$$

where $r_A$ denotes the distance between atom A and the reference particle, $p_i$ denotes the probability of the reference particle being in state $i$, and $\rho_i$ denotes the average density of state $i$.

**Analysis of Li-ion conduction.** We first compute the expected mean-squared-displacement of each transition at different $t$ using the Bayesian rule,

$$\mathbb{E}[d^2(t)|i \rightarrow j] = \frac{\sum_{t'} d^2(t', t'+t) p_i(t') p_j(t'+t)}{\sum_{t'} p_i(t') p_j(t'+t)}, \tag{13}$$

where $p_i(t)$ is the probability of state $i$ at time $t$, and $d^2(t', t'+t)$ is the mean-squared-displacement between $t'$ and $t'+t$. Then, the diffusion coefficient of each transition $D_{i \rightarrow j}(\tau)$ at the lag time $\tau$ can be calculated by,

$$D_{ij}(\tau) = \frac{1}{6} \frac{\mathrm{d}\mathbb{E}[d^2(t)|i \rightarrow j]}{\mathrm{d}t}\bigg|_{t=\tau}, \tag{14}$$

which is shown in Supplementary Table 2.

Finally, we compute the contribution of each transition to Li-ion conduction with Koopman matrix $\boldsymbol{K}(\tau)$ using the cluster Nernst-Einstein equation[67],

$$\sigma_{ij} = \frac{e^2 N_{Li}}{V k_B T} \pi_i z_{ij} K_{ij}(\tau) D_{ij}(\tau), \tag{15}$$

where $e$ is the elementary charge, $k_B$ is the Boltzmann constant, $V$, $T$ are the volume and temperature of the system, $N_{Li}$ is the number of Li-ions, $\pi_i$ is the stationary distribution population of state $i$, and $z_{ij}$ is the averaged charge of state $i$ and state $j$. The percentage contribution is computed by,

$$\frac{\sigma_{ij}}{\sum_{i,j} \sigma_{ij}}. \tag{16}$$

**Lithium diffusion in the FCC lattice toy system.** The molecular dynamics simulations are performed using the Large-scale Atomic/Molecular Massively Parallel Simulator (LAMMPS)[68], as implemented in the MedeA®[69] simulation environment. A purely repulsive interatomic potential in the form of a Born–Mayer term was used to describe the interactions between Li particles and the S sublattice, while all other interactions (Li–Li and S–S) are ignored. The cubic unit cell includes one Li atom and four S atoms, with a lattice parameter of 6.5 Å, a large value allowing for a low energy barrier. 200 ns MD simulations are run in the canonical ensemble (nVT) at a temperature of 64 K, using a timestep of 1 fs, with the S particles frozen. The atomic positions, which constituted the only data provided to the GDyNet and VAMPnet models, are sampled every 0.1 ps. In addition, the energy following the Tet-Oct-Tet migration path was obtained from static simulations by inserting Li particles on a grid.

**Silicon dynamics at solid–liquid interface.** The molecular dynamics simulation for the Si–Au binary system was carried out in LAMMPS[68], using the modified embedded-atom method interatomic potential[27,28]. A sandwich like initial configuration was created, where Si–Au liquid alloy was placed in the middle, contacting with two {110} orientated crystalline Si thin films. 25 ns MD simulations are run in the canonical ensemble (nVT) at the eutectic point (629 K, 23.4% Si[27]), using a time step of 1 fs. The atomic positions, which constituted the only data provided to the GDyNet model, are sampled every 20 ps.

**Scaling of the algorithm.** The scaling of the GDyNet algorithm is $\mathcal{O}(NMK)$, where $N$ is the number of atoms in the simulation box, $M$ is the number of neighbors used in graph construction, and $K$ is the depth of the neural network.

## Data availability

The MD simulation trajectories of the toy system, the Si–Au binary system, and the PEO/LiTFSI system are available at https://doi.org/10.24435/materialscloud:2019.0017/v1.

## Code availability

GDyNets is implemented using TensorFlow[70] and the code for the VAMP loss function is modified on top of ref. [13]. The code is available from https://github.com/txie-93/gdynet.

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

## Acknowledgements

This work was supported by Toyota Research Institute. Computational support was provided by Google Cloud, the National Energy Research Scientific Computing Center, a DOE Office of Science User Facility supported by the Office of Science of the U.S. Department of Energy under Contract No. DE-AC02-05CH11231, and the Extreme Science and Engineering Discovery Environment, supported by National Science Foundation grant number ACI-1053575.

## Author contributions

T.X. developed the software and performed the analysis. A.F.-L. and Y.W. performed the molecular dynamics simulations. T.X., A.F.-L., Y.W., Y.S.H., and J.C.G. contributed to the interpretation of the results. T.X. and J.C.G. conceived the idea and approach presented in this work. All authors contributed to the writing of the paper.

## Additional information

**Competing interests:** The authors declare no competing interests.

