## [Peer Review File · Nature Communications]

Reviewer #1 (Remarks to the Author):

The manuscript by Grossman and co-workers presents a timely approach for the study of complex dynamics in atomistic systems. Furthermore, it is a nontrivial and significant improvement on the (cited) similar work by Noe' and co-workers. The presented application examples, in increasing order of complexity and relevance are well chosen.

We thank the reviewer for his/her careful evaluations of our manuscript and recommendation to publish this work in Nature Communications after some revisions. We are pleased to see that the reviewer comments that our work is “nontrivial” and “significant improvement”, and praises that our examples are “well chosen”.

In the revised manuscript, we have addressed all the comments raised by the reviewer. We are especially thankful for the reviewer’s suggestion to rethink the introduction. We believe that the overall quality and clarity of the manuscript is improved as a result. Below, we provide a concise reply to each comment.

Before recommending publication of this paper in Nature Communications, though, I would like to authors to explain

- how bonds are defined when building the connectivity graph

We thank the reviewer for pointing out this unclarity. The bonds are defined using the nearest neighbor algorithm. For each atom, we connect it with its M nearest neighbors while constructing the graph. At the same time, a special gated architecture is used in GCN to reweigh the strength of each connection. This gated architecture was one key innovation from our previous work [Phys. Rev. Lett. 120, 145301] that allows us to use this relatively simple graph construction algorithm. We have revised our manuscript to include this additional detail. In the paragraph starting “Learning feature map function”, we added the following sentence in the revised manuscript.

We connect M nearest neighbors considering periodic boundary condition while constructing the graph, and a special gated architecture is used in GCN to reweigh the strength of each connection [18].

Also, we added information about the number of M used for constructing the graphs in each system in the methods section.

The number of nearest neighbors M is 12, 20, and 20 for the toy system, Si-Au binary system, and PEO/LiTFSI system, respectively.

- crucially, how the number of states / dimension of the feature space is selected. Is it optimized in some sense, meaning that there are two competing aspects that are balanced at an optimum dimension?

The number of states / dimension is selected using hyperparameter optimization. There are two different aspects here: 1) in terms of overfitting and underfitting, there exists an optimum number of states using the VAMP-2 score as the validation score; 2) in terms of understanding the learned model, a smaller number of states is preferable. Usually, the optimum number from 1) is too large for us to understand the most important dynamics in the system. So, we select the number of states by choosing the smallest number where the CK equation can still be fulfilled within statistical uncertainty. This is described in detail in the section of “Hyperparameter optimization and model validation”. To make this clearer, we added the following sentence in the revised manuscript.

In general, increasing the dimension of feature space makes the dynamic model more accurate, but it may result in overfitting when the dimension is very large.

The next sentence in the original manuscript is also helpful to explain how we select the number of states.

Since a higher feature space dimension and a larger τ make the model harder to understand and contain less dynamical details, we select the smallest feature space dimension and τ that fulfills the CK equation within statistical uncertainty.

I would also suggest to rethink the introduction to make it more appealing for the nonexpert reader. At present it seems a story for initiated related to Koopman analysis and VAMPnets, which, I am sure, tell nothing to most of the readers. A practical example, e.g., by anticipating some to the results, might clarify why the presented approach is interesting to a broad audience.

We appreciate this helpful suggestion to help us make the manuscript more appealing for the nonexpert reader. We have significantly revised the introduction section by adding the following sentences in the paragraph starting “The advances in applying”.

Ideally, we want to trace every atom or small molecule of interest in the MD trajectories, and summarize their dynamics into a linear, low dimensional model that describes how their local environments evolve over time. Recent studies show that combining Koopman analysis and deep neural networks provides a powerful tool to understand complex biological processes and fluid dynamics from data. [11–13] In particular, VAMPnets [13] develop a variational approach for Markov processes to learn a linear model which encodes the long-time dynamics from MD data. However, in order to learn the atomic dynamics in complex, multi-component material systems, sharing knowledge learned for similar local chemical environments is essential to reduce the amount of data needed.

A minor comment: the authors should explain a bit what the lag time vs timescales plots mean in the three cases, including what the color of the curves are referred to.

Thanks for pointing this out. Since the explanation is slightly lengthy and technical, we added the following paragraph in the supplemental information of the manuscript.

Determination of the relaxation timescales. The relaxation timescales represent the characteristic timescales implied by the transition matrix $K(\tau)$, where τ denotes the lag time of the transition matrix. By conducting an eigenvalue decomposition for $K(\tau)$, we could compute the relaxation timescales as a function of lag time by,

$$t_i(\tau) = -\frac{\tau}{\ln|\lambda_i(\tau)|}$$

where $\lambda_i(\tau)$ denotes the i th eigenvalue of the transition matrix K . These $t_i(\tau)$ are plotted in Figs. 2(b), 3(d), and 4(d) for each material system. If the dynamics of the system is Markovian, i.e. Eq. 6 holds, one can prove that the relaxation timescales $t_i(\tau)$ will be constant for any τ . Therefore, we select a smallest τ^* from Figs. 2(b), 3(d), and 4(d) to obtain a dynamical model that is Markovian and contains most dynamical details. We then compute the relaxation timescales using this τ^* for each material system, and these timescales remain constant for any $\tau > \tau^*$.

Reviewer #2 (Remarks to the Author):

Authors suggest a Graph Dynamical Networks method to study the dynamics of molecular systems. The method is built as an extension of VAMPnets, ref 13. Overall this work is rather technical, does not provide a clear breakthrough advantage vs other methods. It is probably more suited to a specialized journal like J Chem Phys, Molecular Simulations or PCCP.

We thank the reviewer for his/her careful evaluation of our manuscript. However, we respectfully disagree with the reviewer's conclusion that our work "is rather technical" and "does not provide a clear breakthrough". As an overall response to the reviewer's comments, we believe that we have achieved the following breakthroughs which justify publication in Nature Communications.

- 1) **Technical breakthrough.** In this work, we propose to use graph convolutional neural networks to encode local chemical structures in VAMPnets. This enables the sharing of knowledge learned for different local chemical environments, which significantly improves the efficiency of learning. As demonstrated with the toy system, our method learns much better dynamics of the lithium atoms than VAMPnets with a fully-connected neural network (FCNN). Furthermore, the silicon liquid-solid interface and polymer electrolyte examples include 8k and 3k atoms in their simulation boxes, respectively. It is not possible to learn dynamics using the VAMPnets with a FCNN when the number of atoms is so large. Our method enables the direct learning of atomic scale dynamics in complex material systems for the first time.
- 2) **Broad applicability.** The method proposed in this work can be applied to a wide range of material systems where the dynamics of individual atoms / molecules are important to material performance. Such material systems include ion transport in batteries, water dynamics in membranes, and gas adsorption in hydron storage materials, to name only a few examples. Also, many different types of materials including inorganic solids, polymers, metal organic frameworks (MOFs) are being actively studied for such applications. In this broad range of fields, MD simulations are widely adopted for understanding atomic scale dynamics, but often only hand-designed features are used currently. We believe that our work can immediately add value to these large amounts of MD data and will be beneficial to a broad material science community.

Based on the reviewer's suggestions, we have carried out additional calculations and revised the manuscript to help clarify our overall contribution to the field. Below, we provide point-to-point responses to the reviewer's comments.

Major comments:

1. No clear head to head comparison with VAMPnets is performed.

We appreciate the reviewer's comment and to make sure it is clear that such a comparison is being made, we have changed the following sentence in the section "Toy system: local vs. global dynamics" to clarify that the same input data is used:

To demonstrate the advantage of learning local dynamics in material systems, we compare the dynamics learned by the GDyNet with VAMP loss and a standard VAMPnet with fully connected neural networks that learns global dynamics for a simple model system using the same input data.

In this comparison, we demonstrated that our method achieves significantly better results by learning local dynamics instead of global dynamics of the lithium atoms as VAMPnets. The learning of the local dynamics is achieved through the parameter sharing and symmetry built in the graph convolutional neural network architecture. We also modify the following sentence to clarify again that the same input data is used.

In contrast, the standard VAMPnet, which inputs the same data as the GDyNet, learns a global transition with a much longer relaxation timescale at 236ps, and it performs much worse in the CK test.

2. Silicon solid-liquid interface is very well studied system. Current simulations should be interpreted and compared with other methods.

We thank the reviewer for his/her suggestion to make addition comparison with existing methods. In the section of "Silicon dynamics in solid-liquid interface", we perform new calculations to compare our results with a q_3 order parameter, which is a classical method to study the liquid-solid structures of silicon. We also want to note that it is not easy to study liquid-solid interfaces in general due to the lack of structural descriptors. We have added additional discussions in the main text and a new Figure S2 in the supplemental information.

These states provide a more detailed description of the solid-liquid interface structure than conventional methods. In Fig. S2, we compare our results with the distribution of the q_3 order parameter of the Si atoms in the system, which measures how much a site deviates from a diamond-like structure and is often used for studying Si interfaces [28]. We learn from the comparison that 1) our method successfully identifies the bulk liquid and solid states, and learns additional interface states that cannot be obtained from q_3 ; 2) the states learned by our method are more robust due to access to dynamical information, while q_3 can be affected by the accidental order structures in the liquid phase; 3) q_3 is system specific and only works for diamond-like structures, but the GDyNets can potentially be applied to any material given the MD data.

FIG. S2. Comparison between the learned states and q_3 order parameters for silicon atoms at the solid-liquid interface. (a) Cross section of the system, where the silicon atoms are color-coded with their q_3 order parameters. (b) Distribution of the q_3 order parameter for the silicon atoms of each state.

3. Overall authors doing poor job comparing the present approach with the state of the art. What are the drawbacks of this approach? If the method is focused on local dynamics how applicable it is to a very slow process (i.e protein folding). Authors should carefully discuss the limitations of their approach.

We appreciate the reviewer's suggestion to do a more thorough comparison with state of the art. We have expanded our previous comparison with the state-of-the-art generative models by including more references and discussing our limitations.

For instance, the auto-encoder architecture [12, 60, 61] and deep generative models [62] start to enable the direct generation of future structures in the configuration space. Our method currently lacks a generative component, but this can potentially be achieved with a proper graph decoder [63, 64].

We have also added more discussions on how our method can potentially be extended to learn the dynamics of protein folding.

In addition, it is easy to extend this method to learn global dynamics with a global pooling function [18]. However, a hierarchical pooling function is potentially needed to directly learn the global dynamics of large biological systems including thousands of atoms.

Finally, we have included more discussion about the limitations of our method.

However, these built-in invariances may also cause the Koopman model to ignore dynamics between symmetrically equivalent structures which might be important to the material

performance. One simple example is the flip of ammonia molecule [59] – the two states are mirror symmetric to each other so the GCN will not be able to differentiate them by design. This can potentially be resolved by partially break the symmetry of GCN based on the symmetry of the material systems.

4. From the ML point of view, this method is not new, rather than the extension of ref 13. But I see no comparison of performance with any baseline. Perhaps VAMPnets is already good, so why bother?

We appreciate the reviewer’s comments and we have revised our manuscript to make it clearer that a comparison is done between our methods and the VAMPnets using the toy system. Our method has the following advantages over the VAMPnets based on the results: 1) our method performs much better in CK tests; 2) by learning the local dynamics of the system, the number of states needed to describe the system is exponentially smaller; 3) global dynamics can be reconstructed from the local dynamics in the toy system.

Furthermore, it is not possible to use the original VAMPnets with fully connected neural networks in the latter two examples in the manuscript. These two systems include thousands of atoms (tens of thousands of degrees of freedom), which is too large for the VAMPnets to learn meaningful dynamics without sharing knowledge between local structures.

Other comments:

“Despite being widely used in biophysics [30] and uid dynamics [38], Koopman models, (or Markov state models [30], master equation methods [39, 40]) have not been used in learning dynamics in materials”

This is not true! Unfortunately, the authors did not bother to search for literature past the field of atomistic simulations. Just a few examples to name

10.1063/1.2959573
10.1016/j.electacta.2005.05.067
10.1155/2013/108386
10.1021/ct900620b
10.1021/jp062693x

We thank the reviewer for pointing us towards these references. We believe these works can be summarized to “kinetic modeling of chemical reactions”, and we have included them in the sentence. We didn’t include 10.1063/1.2959573 and 10.1021/ct900620b because we think they are closer to pure method development works. Also, we clarified our statement to limit our claim to “learning atomic scale dynamics in materials from MD simulations”.

Despite being widely used in biophysics [31], fluid dynamics [39], and kinetic modeling of chemical reactions [40–42], Koopman models, (or Markov state models [31], master equation

methods [43, 44]) have not been used in learning atomic scale dynamics in materials from MD simulations except for a few examples in understanding solvent dynamics [45–47].

The convolutional neural network is a 2d method, essentially neglecting 3d spatial orientation of atoms. How this is affecting the accuracy of simulations/analysis.

We thank the reviewer for pointing out this unclarity. In fact, our graph convolutional neural network is a 3d method, which is accomplished by including the bond length in the graph construction process. The 3d spatial information is extremely important for correctly encoding the local chemical environments of each atom. We have added the following sentence to make this point clearer.

Also, the 3-dimensional information is preserved in the graphs since the bond length is encoded in $u_{i,j}$.

The title is a bit misleading “Unsupervised Learning of Atomic Scale Dynamics in Materials”, since authors use a loss function and minimize it, it’s not purely unsupervised anymore.

A classical definition of unsupervised learning is that only a set of inputs x is used in the learning process. In contrast, in supervised learning, both inputs x and their corresponding labels y are used. In our case, only the coordinates of the atoms are used as inputs to the model without the use of any labels. We believe that our method should be classified into a subclass of unsupervised learning called “dimensionality reduction”, where the high dimensional configuration space is reduced to lower dimensional representations in the state space.

In our opinion, the use of loss is not the criteria for determining whether an algorithm is supervised or unsupervised. Many unsupervised learning algorithms use a loss function as well. For example, the K-means clustering algorithm minimizes the following loss function.

$$\sum_{i=0}^n \min_{\mu_j \in C} (\|x_i - \mu_j\|^2)$$

We appreciate the reviewer’s comments, but we believe the term “unsupervised learning” is warranted to distinguish our work from several recent works that employ supervised learning techniques with MD data, where human designed labels are used. [10.1038/nphys3644]

Reviewer #3 (Remarks to the Author):

The authors propose to reconstruct the dynamic behavior of individual atoms in material systems using Koopman theory and machine learning. Specifically, they recover a Markov state model for the molecular dynamics process of material systems. A linearization of this non-linear process is enabled via embedding of the atomic configuration using a neural network.

The main methodical contribution of this work regards the NN used to generate the representation of the Markov state (i.e. the network architecture used for the embedding): instead of a global representation of the whole system at once, the authors propose a fragmentation into smaller atom-centered neighborhoods. From local atomic representations, a more efficient sampling of the dynamical process is gained due to the structural regularity of materials, because similar local neighborhoods are reoccurring within the same structure. The proposed neural network (NN) directly outputs the feature embedding of the atomic configuration and includes all relevant invariances, making it particularly efficient.

The authors use a recent method “VAMPnets” by Mardt and Pasquali et al. [1] but modify the latent state embedding NN in a way that introduces parameter sharing. The key idea behind VAMPnets, including its cost function (“VAMP score”) is adopted (with attribution) from the original paper. I would like to remark that the development of VAMPnets arises from a non-trivial reformulation of the dynamical system recovery problem by the original authors. On these grounds, my only issue with the manuscript is that the presentation of the proposed “GDyNet” method gives the wrong impression that GDyNet changes the fundamental principles behind VAMPnet. Instead, the proposed method is a specialization that enables handling of structured systems with high data efficiency, through a modified embedding scheme.

My main request for the authors is therefore to include the term “VAMPnets” in their method name to acknowledge the fact that they are developing a variation of it (e.g. "Graph-VAMPnets" or "GC-VAMPnets”, etc.). This should also be clarified in the text. In particular the following statements from the paper are problematic, as they condense the contributions of VAMPnets too strongly:

- “For example, VAMPnets [13] streamlined the process of building Markov state models, a technique for modeling protein folding kinetics from MD data”
- "Our method differs from previous techniques like VAMPnets as we focus on the modeling of local atomic dynamics instead of global dynamics."

A fair introduction to VAMPnets could look something like this: "VAMPnets [1] employ the variational approach for Markov processes in order to learn an optimal latent space representation which encodes the long-time dynamics, and use that to obtain a linear dynamical model between two subsequent time steps x_t , $x_{t+\tau}$. While [1] employed dense neural networks to learn the latent space encoding, here we develop [*name*, e.g. CG-VAMPnets] which employs graph convolution networks that encode important invariances in

the data and allow us to use parameter sharing between different local chemical environments of the system."

The authors document a series of well-designed numerical experiments performed with the reformulated VAMPnet-model. On a toy system, the newly proposed model is compared to the original VAMPnet, demonstrating significant improvements in that scenario. Additionally the authors study two larger systems to investigate whether their method is able to recover complex local structures formed during phase transitions, which it successfully does.

Altogether, the manuscript is well written, with a clear structure and easy to follow flow of arguments. The proposed method improves on existing work in a significant way, which is convincingly demonstrated using several numerical experiments of varying degree of complexity. The experiments are well analyzed. Although the contributions of this paper are put into context of immediately related existing work, I think that the manuscript would benefit from a wider overview of the field of ML-based force fields for MD simulations.

We appreciate the reviewer for his/her careful, detailed review of our manuscript and helpful suggestions to clarify the contributions of VAMPnets. We deeply appreciate the work of the VAMPnets and certainly had no intention to condense its contribution. In the revised introduction part, we adopted the reviewer's suggestion to further highlight the importance of the variational loss function developed by the VAMPnets.

In particular, VAMPnets [13] develop a variational approach for Markov processes to learn an optimal latent space representation that encodes the long-time dynamics, which enables the end-to-end learning of a linear dynamical model directly from MD data.

At the same time, we have made it very clear that we adopted the loss function from the VAMPnets directly.

In this work, we develop a deep learning architecture, Graph Dynamical Networks (GDyNets), that combines Koopman analysis and graph convolutional neural networks to learn the dynamics of individual atoms in material systems. The graph convolutional neural networks allow for the sharing of knowledge learned for similar local environments across the system, and the variational loss developed in the VAMPnets is employed to learn a linear model for atomic dynamics.

We would like to keep the name, "graph dynamical networks", because we believe the major contribution in this manuscript is to demonstrate that sharing the knowledge learned for similar local environments can significantly improve the sampling of atomic dynamics in material systems. This idea is not just limited to using the VAMP loss but rather can be broader. In the discussion section, we make this clearer and mention several directions to further improve this method by incorporating recent ideas in the field of Koopman analysis.

The graph dynamical networks can be further improved by incorporating ideas from both the fields of Koopman models and graph neural networks. For instance, auto-encoder architecture [12, 60, 61] and deep generative models [62] start to enable the direct generation of future structures in the configuration space. Our method currently lacks a generative component, but this can potentially be achieved with a proper graph decoder [63, 64]. Furthermore, transfer learning on graph embeddings may reduce the number of MD trajectories needed for learning the dynamics [65, 66].

In addition, in order to continue to make sure more credit is attributed to VAMP, we have highlighted the use of VAMP loss each time in the manuscript, whenever we discuss the results of a new example.

To demonstrate the advantage of learning local dynamics in material systems, we compare the dynamics learned by the GDyNet with VAMP loss and a standard VAMPnet with fully connected neural networks that learns global dynamics for a simple model system using the same input data.

To evaluate performance of the GDyNets with VAMP loss for a more complicated system, we study the dynamics of silicon atoms at a binary solid-liquid interface.

Finally, we apply GDyNets with VAMP loss to study the dynamics of lithium ions (Li-ions) in solid polymer electrolytes (SPEs), an amorphous material system composed of multiple chemical species.

Here is an additional suggestion:

- in Fig. 1, "dynamic loss" should be replaced by VAMP loss.

We have revised Fig. 1 according to the reviewer's suggestions.

FIG. 1. Illustration of the graph dynamical VAMPnets architecture. The MD trajectories are represented by a series of graphs dynamically constructed at each time step. The red nodes denote the target atoms whose dynamics we are interested in, and the blue nodes denote the rest of the atoms. The graphs are input to the same graph convolutional neural network to learn an embedding $v_i^{(K)}$ for each atom that represents its local configuration. The embeddings of the target atoms at t and $t + \tau$ are merged to compute a VAMP loss that minimizes the errors in Eq. 2 [13, 24].

All in all, I recommend to address these minor shortcomings before publication.

[1] Mardt, A., Pasquali, L., Wu, H., & Noé, F. (2018). VAMPnets for deep learning of molecular kinetics. Nature Communications, 9(1), 5.

Stefan Chmiela

Again, we thank the reviewer for recommending the publication of our work in Nature Communications after minor revisions.

Reviewers' comments:

Reviewer #1 (Remarks to the Author):

The manuscript by Grossman and co-workers presents a timely approach for the study of complex dynamics in atomistic systems. Furthermore, it is a nontrivial and significant improvement on the (cited) similar work by Noe' and co-workers. The presented application examples, in increasing order of complexity and relevance are well chosen.

Before recommending publication of this paper in Nature Communications, though, I would like to authors to explain

- how bonds are defined when building the connectivity graph
- crucially, how the number of states / dimension of the feature space is selected. Is it optimized in some sense, meaning that there are two competing aspects that are balanced at an optimum dimension?

I would also suggest to rethink the introduction to make it more appealing for the nonexpert reader. At present it seems a story for initiated related to Koopman analysis and VAMPnets, which, I am sure, tell nothing to most of the readers. A practical example, e.g., by anticipating some to the results, might clarify why the presented approach is interesting to a broad audience.

A minor comment: the authors should explain a bit what the lag time vs timescales plots mean in the three cases, including what the color of the curves are referred to.

Reviewer #2 (Remarks to the Author):

Authors suggest a Graph Dynamical Networks method to study the dynamics of molecular systems. The method is built as an extension of VAMPnets, ref 13. Overall this work is rather technical, does not provide a clear breakthrough advantage vs other methods. It is probably more suited to a specialized journal like J Chem Phys, Molecular Simulations or PCCP.

Major comments:

1. No clear head to head comparison with VAMPnets is performed.
2. Silicon solid-liquid interface is very well studied system. Current simulations should be interpreted and compared with other methods.
3. Overall authors doing poor job comparing the present approach with the state of the art. What are the drawbacks of this approach? If the method is focused on local dynamics how applicable it is to a very slow process (i.e protein folding). Authors should carefully discuss the limitations of their approach.
4. From the ML point of view, this method is not new, rather than the extension of ref 13. But I see no comparison of performance with any baseline. Perhaps VAMPnets is already good, so why bother?

Other comments:

"Despite being widely used in biophysics [30] and uid dynamics [38], Koopman models, (or Markov state models [30], master equation methods [39, 40]) have not been used in learning dynamics in materials"

This is not true! Unfortunately, the authors did not bother to search for literature past the field of atomistic simulations. Just a few examples to name

10.1063/1.2959573
10.1016/j.electacta.2005.05.067
10.1155/2013/108386
10.1021/ct900620b
10.1021/jp062693x

The convolutional neural network is a 2d method, essentially neglecting 3d spatial orientation of atoms. How this is affecting the accuracy of simulations/analysis.

The title is a bit misleading "Unsupervised Learning of Atomic Scale Dynamics in Materials", since authors use a loss function and minimize it, it's not purely unsupervised anymore.

Reviewer #3 (Remarks to the Author):

The authors propose to reconstruct the dynamic behavior of individual atoms in material systems using Koopman theory and machine learning. Specifically, they recover a Markov state model for the molecular dynamics process of material systems. A linearization of this non-linear process is enabled via embedding of the atomic configuration using a neural network.

The main methodical contribution of this work regards the NN used to generate the representation of the Markov state (i.e. the network architecture used for the embedding): instead of a global representation of the whole system at once, the authors propose a fragmentation into smaller atom-centered neighborhoods. From local atomic representations, a more efficient sampling of the dynamical process is gained due to the structural regularity of materials, because similar local neighborhoods are reoccurring within the same structure. The proposed neural network (NN) directly outputs the feature embedding of the atomic configuration and includes all relevant invariances, making it particularly efficient.

The authors use a recent method "VAMPnets" by Mardt and Pasquali et al. [1] but modify the latent state embedding NN in a way that introduces parameter sharing. The key idea behind VAMPnets, including its cost function ("VAMP score") is adopted (with attribution) from the original paper. I would like to remark that the development of VAMPnets arises from a non-trivial reformulation of the dynamical system recovery problem by the original authors. On these grounds, my only issue with the manuscript is that the presentation of the proposed "GDyNet" method gives the wrong impression that GDyNet changes the fundamental principles behind VAMPnet. Instead, the proposed method is a specialization that enables handling of structured systems with high data efficiency, through a modified embedding scheme.

My main request for the authors is therefore to include the term "VAMPnets" in their method name to acknowledge the fact that they are developing a variation of it (e.g. "Graph-VAMPnets" or "GC-VAMPnets", etc.). This should also be clarified in the text. In particular the following statements from the paper are problematic, as they condense the contributions of VAMPnets too strongly:

- "For example, VAMPnets [13] streamlined the process of building Markov state models, a technique for modeling protein folding kinetics from MD data"
- "Our method differs from previous techniques like VAMPnets as we focus on the modeling of local atomic dynamics instead of global dynamics."

A fair introduction to VAMPnets could look something like this: "VAMPnets [1] employ the variational approach for Markov processes in order to learn an optimal latent space representation which encodes the long-time dynamics, and use that to obtain a linear dynamical model between two subsequent time steps x_t , $x_{t+\tau}$. While [1] employed dense neural networks to learn the

latent space encoding, here we develop [*name*, e.g. CG-VAMPnets] which employs graph convolution networks that encode important invariances in the data and allow us to use parameter sharing between different local chemical environments of the system."

The authors document a series of well-designed numerical experiments performed with the reformulated VAMPnet-model. On a toy system, the newly proposed model is compared to the original VAMPnet, demonstrating significant improvements in that scenario. Additionally the authors study two larger systems to investigate whether their method is able to recover complex local structures formed during phase transitions, which it successfully does.

Altogether, the manuscript is well written, with a clear structure and easy to follow flow of arguments. The proposed method improves on existing work in a significant way, which is convincingly demonstrated using several numerical experiments of varying degree of complexity. The experiments are well analyzed. Although the contributions of this paper are put into context of immediately related existing work, I think that the manuscript would benefit from a wider overview of the field of ML-based force fields for MD simulations.

Here is an additional suggestion:

- in Fig. 1, "dynamic loss" should be replaced by VAMP loss.

All in all, I recommend to address these minor shortcomings before publication.

[1] Mardt, A., Pasquali, L., Wu, H., & Noé, F. (2018). VAMPnets for deep learning of molecular kinetics. *Nature Communications*, 9(1), 5.

Stefan Chmiela

REVIEWERS' COMMENTS:

Reviewer #1 (Remarks to the Author):

Authors' replies to my comments and related changes in the manuscript (or Suppl. Mat.) are satisfactory.

I stand of my opinion that this manuscript is suited for Nat. Comm.

I would ask the author to consider a further clarification (in other words: optional revision), related to my previous comments, on the definition of bond connectivity.

I believe readers should have as much as possible all the needed info for reproducing the results in the manuscript + SM, without percolating through literature to find bits and pieces. Specifically, I noted that the wording "gated architecture" is never used in PRL 2018 (nor in JCP 2018 by the same two authors).

I find this wording only slides of a talk given by Tian Xie (<http://txie.me/assets/pdfs/talks/2018/MRS-CGCNN.pdf>)

"Gating" in graph CNNs is indeed a used wording, but, it seems to me, not always in the same way. I believe some more tutorial explanation, e.g., in the SM, would only increase the impact of the paper.

Reviewer #2 (Remarks to the Author):

Agree with the revision.

Reviewer #3 (Remarks to the Author):

The revised manuscript satisfactorily addresses the minor shortcomings I have raised in my original review. Accordingly, I believe the present version is suitable for publication in Nature Communications

Reviewer #1 (Remarks to the Author):

Authors' replies to my comments and related changes in the manuscript (or Suppl. Mat.) are satisfactory.

I stand of my opinion that this manuscript is suited for Nat. Comm.

I would ask the author to consider a further clarification (in other words: optional revision), related to my previous comments, on the definition of bond connectivity.

I believe readers should have as much as possible all the needed info for reproducing the results in the manuscript + SM, without percolating through literature to find bits and pieces. Specifically, I noted that the wording "gated architecture" is never used in PRL 2018 (nor in JCP 2018 by the same two authors).

I find this wording only slides of a talk given by Tian Xie (<http://txie.me/assets/pdfs/talks/2018/MRS-CGCNN.pdf>)

"Gating" in graph CNNs is indeed a used wording, but, it seems to me, not always in the same way. I believe some more tutorial explanation, e.g., in the SM, would only increase the impact of the paper.

We thank the reviewer for his/her careful read and helpful suggestions. In the revised manuscript, we have added additional explanations about the "gated architecture" in the supplementary information.

Supplementary Note 1: gated architecture in graph convolutional neural networks. The gated architecture in GCN is important for learning the local environments in a complex material system. In general, it is challenging to define bonds, i.e. the topology of the graph, in materials to capture the non-covalent interactions which may affect the atomic dynamics. We resolve this challenge by introducing the gated architecture in GCN to reweigh the strength of each connection. In the graph construction, we connect each atom with its M nearest neighbors, where the M is large enough to include non-covalent interactions. During the training, the gated architecture (or the attention layer as described in the methods section) automatically learns a weight factor that is related to the bond length and atom types. This weight factor reweighs the importance of the M nearest neighbors to center atom. As a result, we can learn a representation of the local environments in complex materials using the graphs constructed by the simple nearest neighbor approach.

Reviewer #2 (Remarks to the Author):

Agree with the revision.

Reviewer #3 (Remarks to the Author):

The revised manuscript satisfactorily addresses the minor shortcomings I have raised in my original review. Accordingly, I believe the present version is suitable for publication in Nature Communications.

Finally, we also included several additional citations that are relevant to our work.

Our approach also differs from several other unsupervised learning methods [48–50] by directly learning a linear Koopman model from MD data.